# Mechanism Design for Large Language Models

## ABSTRACT

We investigate auction mechanisms to support the emerging format of AI-generated content. We in particular study how to aggregate several LLMs in an incentive compatible manner. In this problem, the preferences of each agent over stochastically generated contents are described/encoded as an LLM. A key motivation is to design an auction format for AI-generated ad creatives to combine inputs from different advertisers. We argue that this problem, while generally falling under the umbrella of mechanism design, has several unique features. We propose a general formalism—the *token auction* model— for studying this problem. A key feature of this model is that it acts on a token-by-token basis and lets LLM agents influence generated contents through single dimensional bids.

We first explore a robust auction design approach, in which all we assume is that agent preferences entail partial orders over outcome distributions. We formulate two natural incentive properties, and show that these are equivalent to a monotonicity condition on distribution aggregation. We also show that for such aggregation functions, it is possible to design a second-price auction, despite the absence of bidder valuation functions. We then move to designing concrete aggregation functions by focusing on specific valuation forms based on KL-divergence, a commonly used loss function in LLM. The welfare-maximizing aggregation rules turn out to be the weighted (log-space) convex combination of the target distributions from all participants. We conclude with experimental results in support of the token auction formulation.

ACM Reference Format:

Anonymous Author(s). 2024. Mechanism Design for Large Language Models. In *Proceedings of ACM Conference (Conference'17)*. ACM, New York, NY, USA, 12 pages. https://doi.org/XXXXXXX.XXXXXXX

## 1 INTRODUCTION

In the current web ecosystem, auctions are the primary mechanism used to decide which ads (and commercial content more broadly) are displayed to users [10, 19]. In those, advertisers bid for the right to have their creatives displayed to the user along with organic contents. Many of the web formats such as text, banners, video, apps, ... have their own subtleties which led to the development of new auction tools to handle them. Our goal in this paper is to investigate auction mechanisms to support the emerging format of AI-generated content. More specifically, we explore the use of auctions as a tool for influencing the output of large language models (LLMs) [e.g., 6].

*Conference'17, July 2017, Washington, DC, USA*
© 2024 Association for Computing Machinery.
ACM ISBN 978-x-xxxx-xxxx-x/YY/MM...$15.00
https://doi.org/XXXXXXX.XXXXXXX

We consider a situation where a certain space in the web (be a webpage, an UI element of an AI-chatbot, the dialog of a certain character in a video or a game ...) is clearly marked as commercial content and different advertisers can bid to influence the content in that space. Each advertiser has an LLM that can generate content for that space and is willing to pay a certain amount of money for the right to have their content displayed. A simple design is to collect bids from advertisers and let the highest bidder choose whatever content they wish to publish in that space. While simple, this design does not exploit the flexibility of LLMs which is to combine different concepts in a creative way.

Consider this example. In the first we ask an LLM to produce different ads for the fictitious Stingray resort and the equally fictitious Maui Airlines:

- *"Experience the magic of Hawaii at Stingray Resort, where stunning views, luxurious accommodations, and endless activities await. Book your stay today and create unforgettable memories in the heart of paradise."*
- *"Fly to Hawaii with Maui Airlines and experience the beauty of the Aloha State. We offer affordable flights to all the major islands, so you can start your Hawaiian vacation sooner. Book your flight today and let the island spirit take over!"*

For that use case, however, the LLM is flexible enough to produce a joint ad for both:

- *"Fly to paradise with Maui Airlines and experience the magic of Hawaii at Stingray Resort. Stunning views, luxurious accommodations, and endless activities await. Book your dream vacation today and create unforgettable memories."*

One can envision an auction mechanism where both Stingray resort and Maui airlines can submit both LLMs as well as bids, and this will determine their prominence in the final outcome.[1]

### 1.1 Unique Challenges

LLMs [1, 6, 18] are a new technology with new and unconventional aspects, many of which have direct implications to auction design (e.g., how preferences are represented/expressed). Our goal is to identify some of the key challenges and take a first step in designing mechanisms to address them.

**Modelling and Expressing Preferences:** Auction theory typically models preferences via value functions that assign a value to each outcome. LLMs, however, are *generative models* which do not attribute values to each example, but instead succinctly encode preferences over outcomes in a stateless neural network model that predicts continuation probabilities.

**Necessity of Randomization:** LLMs crucially rely on randomization. When forced to output deterministically, LLMs typically have a worse performance than if they are allowed to sample from

---

[1]While our main focus is to create ad creatives that merge content from different advertisers, auction mechanisms for merging LLM outputs could be used in other contexts.

a distribution. Therefore, an auction that aggregates LLM outputs must also output distributions.

**Technical Alignment:** Auction solutions should be technically aligned with current LLM technology. They should only use information available from current models and should be easy to integrate in the system. Ideally the allocation and payments should be obtained from simple manipulations of the LLM outputs.

**Computational Efficiency:** LLM models are expensive to query, so the auction computation should not add too much overhead. In particular, auctions should not increase the number of calls to inference the models beyond the minimum necessary.

## 1.2 Our Contributions

**The Token Auction Model.** Our first contribution is a formalism ("The Token Auction Model") for studying this problem. *Tokens* are the units making up sentences and paragraphs.[2] Examples of tokens include (sub-)words, symbols, numbers, and special tokens indicating the beginning and ending of the text. In particular, any piece of text (potentially incomplete) can be represented as an array of tokens, and any array of tokens also encodes a piece of text.

One salient feature of the state-of-the-art LLMs is that they are stateless, i.e., they maintain no internal memory or state. Instead, they simply map a prefix string to a distribution over the next token. The output is then created in an autoregressive manner. Given an input prompt, the output is generated by repeatedly feeding the current sequence of tokens into the LLM, sampling a continuation token, and appending it to the sequence of tokens.

The *token auction* we propose also operates on a token-by-token basis, and serves to aggregate several LLMs to generate a joint output. We assume the designer has access to algorithmic LLM agents represented by their respective text generation functions (the functions that map a sequence of tokens to a distribution over the next token). In addition, we allow each LLM agent to submit a single dimensional bid. The auction output will be an aggregated distribution together with a payment rule.

This approach may appear counterintuitive at first since an advertiser cares about the generated final text, not the specific choice of words. This seems to suggest a dynamic planning of the generated token sequence. However, existing LLMs do not reason about full pieces of text, nor do they plan ahead; instead, their preferences are expressed as desired distributions over merely the next token. In other terms, we can think of an LLM as a succinct distillation of an agent's complex combinatorial preferences over sequences of tokens into a generative token-by-token model.[3]

The problem of aggregating LLMs forces the designer to understand the preferences of the agents away from the distilled LLM. This appears to be a very difficult problem. Specifically, we believe it is implausible or at least impractical to assume an individual agent can meaningfully manipulate the distribution over tokens at any given stage, to direct the produced text to a more preferred one. Our auction formulation seeks to strike a balance: By truthfully

revealing the LLM to the designer, the agent gives the auction mechanism a hint as to what their preferred distribution is. The bids, in turn, can be used to tradeoff between agents, and in particular help the designer determine their relative weights.

**Simple and Robust Token Auctions.** Motivated by the very challenging problem of modelling the agents' preferences for nearby generative models, we aim to design *robust* token auctions. We seek auctions that provide desirable incentive properties, while imposing minimal assumptions on the agents' preferences over distributions.

Specifically, we investigate a model where agents' preferences entail partial orders over distributions. We formulate two desirable incentive properties, which we consider minimal requirements:

- *Payment monotonicity*: Given two different bids for the same agent, a final distribution is closer to the desired distribution if and only if the payment is higher.
- *Consistent aggregation:* If for two different bids of the same agent, the final distribution is closer to the preferred distribution for some bids of the other agents, then it should be so for all bids of the other agents.

We show that any mechanism with these two properties is *strategically equivalent* to a mechanism that satisfies a monotonicity requirement on the distribution aggregation function.

We then investigate whether it is possible to equip such distribution aggregation functions with payment rules that satisfy additional incentive properties. Specifically, we investigate whether such aggregation rules admit an analogue of the *second-price payment rule*. In the single-item second-price (or Vickrey) auction [20], the payment corresponds to the critical bid where an agent transitions from losing to winning. To port this notion to our setting, we show that any monotone aggregation rule can be written as a distribution over deterministic allocations from bids to tokens such that there is a critical bid where the allocation transitions from a less preferred to a more preferred token. Such a critical bid becomes then a natural candidate for a payment rule.

A natural analogue of the second-price auction is obtained for a model that only relies on ordinal preferences. The resulting class of auctions is applicable whenever the agent valuations are compatible with the partial order, and provides robust incentives for all of these.

**Designing Aggregation Functions.** We then move to designing concrete aggregation functions. Our approach is to define welfare objectives inspired by LLM training, and to derive optimal distribution aggregation functions for these welfare notions.

We focus on specific valuation forms based on KL-divergence, a commonly used loss function in LLM. We specifically discuss two natural welfare notions, and show that the corresponding welfare-maximizing aggregation rules are the weighted (log-space) convex combination of the target distributions from all participants.

The linear and log-linear aggregation rules we identify have different pros and cons. Both share the advantage that they are welfare maximizing for the respective welfare notions. The linear rule unlike the log-linear rule is also monotone, and therefore compatible with the robust incentives approach.

**Demonstration.** We conclude with experiments to support our token auction formulation, obtained by prompt-tuning of a publicly available LLM. We consider a two-advertiser example and the linear and log-linear aggregation rules identified to be optimal for

---

[2]More generally, one can consider tokens forming parts of images [14, 25] and videos [17]. For the purpose of this paper, we will stick with text generation.
[3]See our discussion in Section 4, and Propositions 4.1 and 4.3 for additional support for the stateless approach.

corresponding welfare notions. We show how the combined output varies as a function of $\lambda = b_1/(b_1+b_2)$, where $b_1$ and $b_2$ are the advertisers' bids. Both approaches lead to meaningful and interpretable text, and smoothly transition from one to another advertiser, with a joint ad produced for intermediate values of $\lambda$.

**Additional Related Work.** Additional related directions include LLM fine-tuning [2, 3, 12, 21], in-context learning [6, 22, 23], mechanism design for public projects [9, 13], and robust mechanism design [4, 5, 8, 15]. See Appendix A for a more detailed discussion.

## 2 PRELIMINARIES

In this section, we first provide an abstraction of typical generative models. For concreteness we adopt a terminology that suits the important LLM use case where the creative is text. We then introduce the basic formalism of the mechanism design problem we study.

### 2.1 Abstraction of Large Language Models

*Large language models* (LLMs) [1, 6, 18] can be abstracted as functions mapping from a partial sentence to the distribution of the next *token* that extends the partial sentence.

Formally, let $T$ be the set of tokens and $\Delta(T)$ be the set of distributions over $T$. Let $T^* = T \cup T^2 \cup \cdots \cup T^K$ denote the set of sequences of tokens, where $K$ is the maximum sequence length that the LLM can handle. Each LLM is modeled as a function $f : T^* \to \Delta(T)$ that maps any sequence of tokens to a distribution over the next token.

*Autoregressive Text Generation.* A *prompt* is an initial set of tokens $s_0 \in T^*$ provided with instructions of what text to generate. An LLM produces a text in response to the prompt by sampling a token $\tau_1 \sim f(s_0)$ and constructing $s_1 = s_0 \oplus \tau_1$ (where $\oplus$ is the operation to append a token to an array). We then repeat the process of $\tau_k \sim f(s_k)$ and $s_{k+1} = s_k \oplus \tau_k$ until a special *end-of-sentence* token is sampled. If at some point the sequence of tokens becomes too long (larger than $K$) we trim $s_{k+1}$ to its length-$K$ suffix.

We remark that LLMs are stateless by design: They keep no internal memory (other than the sequence of tokens produced so far) and each token is sampled independently.

*Training of LLMs.* An LLM $f$ is parameterized by a neural network structure $M$ and a set of weights $W$. The weights are often obtained by three stages of optimization (see first three rows in Table 1). The initial stage is very computationally intensive but task independent. Subsequent stages are less costly and their goal is to adapt the general purposed model obtained in the first stage to more specific tasks. In each of the stages, we minimize a different loss function over a different dataset. The details of the training process are not particularly relevant for our discussion, but we add a more detailed discussion in Section 4.1. We will note, however, that some of the mechanisms we discuss for combining the inputs of different LLMs resemble the functional form used in the reinforcement learning and fine-tuning steps.

### 2.2 Token Auctions for LLMs

We now formalize the mechanism design problem of combining the outputs of different LLM-represented algorithmic agents. As discussed in the introduction, we will design an auction to act on the token-by-token generation stage. Our goal is to keep the auction technically aligned with the state-of-the-art LLM systems.

*Robust Modeling of LLM Agents' Preferences.* One major challenge in designing mechanisms for LLM agents is that they are represented as distributions, and it is generally difficult to compare LLM agents' "utilities" among distributions. To illustrate, suppose an LLM agent's preferred distribution over two tokens is $p = (0.6, 0.4)$, and consider two possible generated distribution outcomes: $q_1 = (0.5, 0.5)$ and $q_2 = (0.8, 0.2)$. Between $q_1$ and $q_2$, it is unclear which one this LLM agent would prefer since while $q_2$ appears more distant from $p$ than $q_1$, it has a higher probability on the first token which appears more preferably by the LLM's distribution $p$.

Despite this incomparability between $q_1$ and $q_2$, it does appear "obvious" that $q_2$ would be less preferred by the LLM than $q_3 = (0.7, 0.3)$. This is because $q_3$ deviates from $p$ along the same directions as $q_2$ for each entry (i.e., both increase or both decrease), but deviates less in terms of the absolute value of deviation.

The above observation illustrates that while it is difficult to model LLM agents' complete preferences over all the generated distributions, it seems plausible to assume a certain *partial order* over the distributions. This motivates us to consider *robust* modeling of LLM agents' preferences, with the following notion of *obvious preference*.

*Definition 2.1 (Obvious Preferences over Distributions).* Consider any LLM agent $i$ with preferred distribution $p_i$, and any two aggregation distribution $q, q' \in \Delta(T)$. We say $q$ is (weakly) *obviously preferred* over $q'$ by agent $i$ — or formally, $q \succeq_i q'$ — if

$$\forall t \in T, \qquad |q(t) - p_i(t)| \leq |q'(t) - p_i(t)| \qquad (1)$$

$$\text{and} \quad (q(t) - p_i(t))(q'(t) - p_i(t)) \geq 0. \qquad (2)$$

Moreover, if $q \neq q'$, $q$ is strictly preferred over $q'$ by $i$, i.e., $q \succ_i q'$.

In other words, $q$ is preferred by $i$ over $q'$ when (1) the deviation of $q$ from $p_i$ is smaller than the deviation of $q'$ from $p_i$ for every entry; and (2) these deviations are along the same direction for every entry. Note that Definition 2.1 only specified a partial ordering among aggregated distributions. Thus it is possible that two distributions are not comparable, i.e., $q \not\succeq_i q'$ and $q' \not\succeq_i q$.

*Token Auctions.* Our goal is to design simple, practical auction mechanisms that work well under minimal assumptions about the agents' private preferences. Specifically, we seek to design *token auction mechanisms* $\mathcal{M} = \langle q, z \rangle$, where $q$ is a distribution aggregation function and $z$ is a payment function. A token auction mechanism operates on a token-by-token basis, and lets $n$ algorithmic LLM agents influence the output distribution and payments through scalar bids. We denote the vector of bids by $\boldsymbol{b} = (b_1, \ldots, b_n) \in \mathbb{R}^n_+$. We assume that the initial prompt $s_0 \in T^*$, and the text aggregation functions $f_1, \ldots, f_n$ of the $n$ LLM agents are publicly known.

*Distribution Aggregation Function.* This is the first ingredient to a token auction mechanism. A distribution aggregation function $q$ takes as input a vector of bids $\boldsymbol{b} \in \mathbb{R}^n_+$ and $n$ distributions $\boldsymbol{p} \in \Delta(T)^n$ and maps these to a distribution over tokens:

$$\text{aggregation function:} \quad q : \mathbb{R}^n_+ \times \Delta(T)^n \to \Delta(T).$$

For fixed bids, a distribution aggregation function can be used in the same way as a text aggregation function. Namely, starting from the initial prompt $s_0 \in T^*$, we can repeatedly sample $\tau_k$ from distribution $q_k = q((b_1, \ldots, b_n), (f_1(s_{k-1}), \ldots, f_n(s_{k-1})))$ for each $k \geq 1$ to generate $s_k = s_{k-1} \oplus \tau_k$. Note the alignment with LLMs,

| Training/Learning stages | Data | Cost | Goal |
|---|---|---|---|
| Pre-training [1, 6] | General texts from web, books, etc | Very high | A common baseline shared across downstream tasks |
| Instruction fine-tuning [21] | Task specific data | Medium | Optimize the behavior for specific tasks |
| RLHF [12] | Human evaluations | Medium | Security control, reducing harmful behavior, etc |
| In-context few-shot learning | Carefully designed prompts as inputs | Very low | Effectively influence the behavior in real-time |

**Table 1: Common training stages of LLMs.**

which already produce the distributions $f_i(s_{k-1})$ for $i \in [n]$. No additional calls to the LLMs are needed.

*Payment Function.* In addition to the distribution aggregation function, we seek to design payment functions. Here, we want to operate on a token-by-token-basis and seek a stage independent design. Formally, for each agent $i$, we aim to define a

$$\text{pricing function:} \quad \zeta_i : \mathbb{R}_+^n \times \Delta(T)^n \times T \to \mathbb{R},$$

with the interpretation that for bids $\boldsymbol{b} \in \mathbb{R}_+^n$, distributions $\boldsymbol{p} \in \Delta(T)^n$, and token $t \sim q(\boldsymbol{b}, \boldsymbol{p})$, the payment from agent $i$ is $\zeta_i(\boldsymbol{b}, \boldsymbol{p}, t)$. These pricing functions naturally lead to expected payments by taking expectations over tokens. Namely, for each agent $i$, we define

$$\text{payment function:} \quad z_i : \mathbb{R}_+^n \times \Delta(T)^n \to \mathbb{R},$$

as the function that takes as input a vector of bids $\boldsymbol{b} \in \mathbb{R}_+^n$ and distributions $\boldsymbol{p} \in \Delta(T)^n$, and maps these to $z_i(\boldsymbol{b}, \boldsymbol{p}) = \mathbb{E}_{t \sim q(\boldsymbol{b}, \boldsymbol{p})}[\zeta_i(\boldsymbol{b}, \boldsymbol{p}, t)]$.

*Discussion.* We believe that token auctions and our assumptions offer the right level of abstraction for reasoning about the strategic aspects of aggregating LLMs. This is because it seems impractical (if not impossible) to fully express an agent's preferences over all possible generated creatives. The most plausible way to do so at the current state of the art is perhaps to represent each agent as an LLM. Indeed, by design, current LLMs naturally distill agent preferences over texts to stateless distributions over tokens. Therefore, if agents' are represented as LLMs, it seems natural to auction tokens based on agents' token preferences expressed by the LLMs.

At the same time, the detailed functioning of LLMs remains rather opaque, and it seems implausible that agents could meaningfully misreport the outcome distributions of their LLMs in order to achieve a more desirable aggregated output.

Our auction formulation offers a middle ground. We assume the designer has access to the LLMs, but let the agents influence the aggregation process through a single dimensional bid.

## 3 INCENTIVES IN TOKEN AUCTIONS

In this section, we examine the strategic properties of token auctions. Our goal is robust incentive properties that rely on as few assumptions about the agents' preferences as possible. We first formulate two natural properties that any reasonable mechanism should satisfy, and show that they are equivalent to a monotonicity requirement on the distribution aggregation function. We then show that for any such monotone distribution aggregation function, it is possible to define a natural second-price payment rule.

## 3.1 Desirable Incentive Properties

We begin by formulating two conditions that any reasonable token auction mechanisms should satisfy. The first is a *monotonicity* condition on the payment function. It requires that agents' pay increases if and only if they obtain obviously better distributions.

*Definition 3.1 (Payment Monotonicity).* Mechanism $\mathcal{M} = \langle q, z \rangle$ satisfies *payment monotonicity*, if for all $\boldsymbol{p}$, $\boldsymbol{b}_{-i}$ and $b_i \geq b_i'$ we have

$$z_i(b_i, \boldsymbol{b}_{-i}, \boldsymbol{p}) \geq z_i(b_i', \boldsymbol{b}_{-i}, \boldsymbol{p}) \iff q(b_i, \boldsymbol{b}_{-i}, \boldsymbol{p}) \succeq_i q(b_i', \boldsymbol{b}_{-i}, \boldsymbol{p}).$$

It is a natural incentive constraint because if the payment function is not monotone, then bidders will naturally manipulate their bids in order to induce better distribution with lower payment.

The second incentive constraint is about the consistency of the aggregation function. Intuitively, whenever two bids lead to two aggregated distributions with an obvious order, this order should be consistent in the sense that it is not affected by others' bids.

*Definition 3.2 (Consistent Aggregation).* The distribution aggregation function $q(\boldsymbol{b}, \boldsymbol{p})$ is said to be *consistent* if it admits consistent ordering across all $\boldsymbol{b}_{-i}$. Formally, if $q(b_i, \boldsymbol{b}_{-i}, \boldsymbol{p}) \succeq_i q(b_i', \boldsymbol{b}_{-i}, \boldsymbol{p})$ for some $\boldsymbol{b}_{-i}$, then for all $\boldsymbol{b}_{-i}'$, $q(b_i, \boldsymbol{b}_{-i}', \boldsymbol{p}) \succeq_i q(b_i', \boldsymbol{b}_{-i}', \boldsymbol{p})$.

Similar to payment monotonicity, this requirement of consistent aggregation is imposed to avoid bidders' concerns that the same bid can lead to obviously better or worse distributions, depending on the opponents' bids. Notably, this constraints only apply to the bids $b_i, b_i'$ such that $q(b_i, \boldsymbol{b}_{-i}, \boldsymbol{p}) \succeq_i q(b_i', \boldsymbol{b}_{-i}, \boldsymbol{p})$ for some $\boldsymbol{b}_{-i}$. If $q(b_i, \boldsymbol{b}_{-i}, \boldsymbol{p})$ and $q(b_i', \boldsymbol{b}_{-i}, \boldsymbol{p})$ are not comparable, then they just remain not comparable under different $\boldsymbol{b}_{-i}'$.

These two properties are quite common in the mechanism design literature. For instance, in single-time auction design, allocation consistency is a necessary condition for incentive compatible mechanisms [11], and it also induces a monotone payment function.

## 3.2 Monotone Aggregation Functions

Next we show a "revelation principle" type of result, stating that if one is interested in mechanisms satisfying the desirable incentive properties stated above (Definition 3.1 and Definition 3.2), then one can without loss of generality focus on *monotone aggregation functions* as captured in the following definition.

*Definition 3.3 (Monotone Aggregation Function).* The distribution aggregation function $q(\boldsymbol{b}, \boldsymbol{p})$ is called *monotone* if any higher bid from any agent $i$ leads to an obviously more preferred aggregated distribution for $i$. Formally, for all $\boldsymbol{p}$, $\boldsymbol{b}_{-i}$ and $b_i \geq b_i'$:

$$q(b_i, \boldsymbol{b}_{-i}, \boldsymbol{p}) \succeq_i q(b_i', \boldsymbol{b}_{-i}, \boldsymbol{p}).$$

We are now ready to state our main finding in this subsection with the following definition of *strategic equivalence* between two mechanisms $\mathcal{M}$ and $\tilde{\mathcal{M}}$. In words, the aggregated distribution and all agents' payments will be the same under mechanism $\mathcal{M}$ and $\tilde{\mathcal{M}}$ after each agent $i$ applies some strategy mapping $\pi_i$. Formally,

*Definition 3.4 (Strategic Equivalence).* Any two mechanisms $\mathcal{M} = \langle q, z \rangle$ and $\tilde{\mathcal{M}} = \langle \tilde{q}, \tilde{z} \rangle$ are *strategically equivalent* if there exists a profile $\pi$ of strategy mappings with a bijection $\pi_i : \mathbb{R}_+ \to \mathbb{R}_+$ for every agent $i$ (i.e., $\pi(\boldsymbol{b}) = (\pi_1(b_1), \ldots, \pi_n(b_n))$), such that $\forall \boldsymbol{b} \in \mathbb{R}_+^n$, $\boldsymbol{p} \in \Delta(T)^n$, $q(\boldsymbol{b}, \boldsymbol{p}) = \tilde{q}(\pi(\boldsymbol{b}), \boldsymbol{p})$ and $z(\boldsymbol{b}, \boldsymbol{p}) = \tilde{z}(\pi(\boldsymbol{b}), \boldsymbol{p})$.

THEOREM 3.5 (REVELATION PRINCIPLE). *Any mechanism $\mathcal{M} = \langle q, z \rangle$ with a consistent distribution aggregation function $q$ and a monotone payment function $z$ is strategically equivalent to a mechanism $\tilde{\mathcal{M}} = \langle \tilde{q}, \tilde{z} \rangle$ which has a monotone distribution aggregation function $\tilde{q}$ and a monotone payment function $\tilde{z}$.*

*Remark. Theorem 3.5 can be viewed as a revelation principle in the sense that it simplifies the design choice of aggregation functions. The monotone aggregation functions are a strict subset of consistent aggregation functions since monotonicity directly implies a total order over possible aggregated distributions $Q(\boldsymbol{b}_{-i}, \boldsymbol{p}) = \{q(b_i, \boldsymbol{b}_{-i}, \boldsymbol{p}) : b_i \in \mathbb{R}_+\}$, with the order naturally determined by the real-numbers' order on $i$'s bid $b_i$ and thus this order will be consistent across different $\boldsymbol{b}_{-i}$ and $\boldsymbol{p}$. In this sense, one might think that consistency — which does not impose any restriction at all whenever $q(b_i, \boldsymbol{b}_{-i}, \boldsymbol{p})$ and $q(b'_i, \boldsymbol{b}_{-i}, \boldsymbol{p})$ are not comparable — might be a significantly weaker requirement on aggregation functions than monotonicity which requires a total and consistent order. Theorem 3.5 shows that this is not the case — they are essentially the same as long as the natural incentive requirement of payment monotonicity is also imposed.*

The proof of Theorem 3.5 hinges on the following two lemmas, Lemma 3.7 and Lemma 3.6. Together these two lemmas imply the existence of a strategy mapping, under which the resulting aggregation function becomes monotone. The proof of the theorem is completed by applying the same mapping to the payment function, and noting that this ensures payment monotonicity. We defer the formal proofs of these results to Appendix B.

LEMMA 3.6. *Consider any consistent distribution aggregation function $q$. Suppose $\succeq_i$ defines a total order over aggregations $Q(\boldsymbol{b}_{-i}, \boldsymbol{p}) = \{q(b_i, \boldsymbol{b}_{-i}, \boldsymbol{p}) : b_i \in \mathbb{R}_+\}$ induced by agent $i$'s bid for any fixed $\boldsymbol{b}_{-i}$ and $\boldsymbol{p}$, then there exist a profile $\pi$ of strategy mappings such that $\tilde{q}(\boldsymbol{b}, \boldsymbol{p}) = q(\pi(\boldsymbol{b}), \boldsymbol{p})$ is a monotone aggregation function.*

LEMMA 3.7. *For any distribution aggregation function $q$, there exists a payment function $z$ such that mechanism $\mathcal{M} = \langle q, z \rangle$ is payment-monotone if and only if $\succeq_i$ establishes a total order over $Q(\boldsymbol{b}_{-i}, \boldsymbol{p}) = \{q(b_i, \boldsymbol{b}_{-i}, \boldsymbol{p}) : b_i \in \mathbb{R}_+\}$ for any fixed $\boldsymbol{b}_{-i}$ and $\boldsymbol{p}$.*

We conclude this subsection by giving two natural examples used in today's machine learning practice: an example of a monotone aggregation function, and a non-monotone one.

*Example 3.8 (Linear Aggregation).* Consider $q_{\mathsf{KL}}(\boldsymbol{b}, \boldsymbol{p})$ defined as
$$q_{\mathsf{KL}} = \frac{1}{B} \sum_{i \in [n]} b_i \cdot p_i, \text{ where } B = \sum_{i \in [n]} b_i.$$
It is easy to verify that this is a monotone aggregation function.

*Example 3.9 (Log-linear Aggregation).* Consider the aggregation function $\bar{q}_{\mathsf{KL}}(\boldsymbol{b}, \boldsymbol{p})$ defined by the following equations:
$$\forall t \in T, \; \ln \bar{q}_{\mathsf{KL}}(t) = \frac{1}{B} \sum_{i \in [n]} b_i \cdot \ln p_i(t) - C,$$
where $B = \sum_{i \in [n]} b_i$ and $C = \ln \sum_{t \in T} e^{\frac{1}{B} \sum_{i \in [n]} b_i \cdot \ln p_i(t)}$.

The following two-agent example shows that $\bar{q}_{\mathsf{KL}}$ is not monotone: $p_1 = (.5, .4, .1)$ and $p_2 = (.5, .1, .4)$. When $b_1 = b_2$, $\bar{q}_{\mathsf{KL}} = (\sqrt{.25}, \sqrt{.04}, \sqrt{.04})/.9 = (5/9, 2/9, 2/9)$. Fix $b_2 = 1$, either $b_1 \to 0$ or $b_1 \to \infty$, $\bar{q}_{\mathsf{KL}}(t_1) = .5 < 5/9$. Hence $\bar{q}_{\mathsf{KL}}$ must not be monotone.

## 3.3 Second Price Payment Rules

Next we explore whether for monotone aggregation functions we can create a pricing rule with a "second-price" semantic inspired by the Vickrey auction notion of "minimum-bid-to-win". In the Vickrey auction, the payment corresponds to the critical bid where an agent transitions from losing to winning. To port this notion to our setting, we will show in Theorem 3.12 that any monotone aggregation rule can be written as a distribution over deterministic allocations from bids to tokens such that there is a critical bid where the allocation transitions from a less preferred to a more preferred token. Such a critical bid becomes then a natural candidate for the payment. We will show that besides being payment monotone it has other desirable properties, such as a Myerson-like characterization [11] in terms of the total variation distance between the preferred distribution and the outcome. To define our payment rule, we first discuss the notion of stable sampling.

*3.3.1 Stable Sampling.* We analyze a monotone distribution aggregation function $q(\boldsymbol{b}, \boldsymbol{p})$ from the perspective of a single agent where the distributions $\boldsymbol{p}$ and the bids of other agents $\boldsymbol{b}_{-i}$ are fixed. To simplify the notation, we refer to $q(b_i, \boldsymbol{b}_{-i}, \boldsymbol{p})$ as $q(b_i)$.

We define an implementation of $q(\cdot)$ as a function $\sigma$ that maps $b_i$ and an exogenous random variable $r \sim \mathcal{R}$ (independent of $\boldsymbol{b}$ and $\boldsymbol{p}$) to a token $t \in T$. When $r$ is fixed, $\sigma$ is fully deterministic. We say that $\sigma : \mathbb{R}_+ \times \mathcal{R} \to T$ is a valid implementation of $q(\cdot)$ if:
$$\Pr_{r \sim R}[\sigma(b_i, r) = t] = q_t(b_i), \forall t \in T.$$

Next we define what it means for an implementation to be a stable sampling. It is useful to split the token set $T$ into $T_+ = \{t \in T : q_t(0) \leq (p_i)_t\}$ and $T_- = \{t \in T : q_t(0) > (p_i)_t\}$ corresponding to undersampled ($T_+$) and oversampled ($T_-$) tokens. The monotonicity of aggregation function $q$ is equivalent to the monotonicity of $q_t(b_i)$ as formalized in the following lemma (proof in Appendix C).

LEMMA 3.10. *A distribution aggregation function $q$ is monotone, if and only if for every agent $i$ and $b_i \in \mathbb{R}_+$,*

(1) *$\forall t \in T_+$, $q_t(b_i) \leq (p_i)_t$ and $q_t$ weakly increases;*
(2) *$\forall t \in T_-$, $q_t(b_i) \geq (p_i)_t$ and $q_t$ weakly decreases.*

We can now define the key notion of stable sampling, and state and prove our main result in this subsection.

*Definition 3.11 (Stable Sampling).* Let $q_i(b_i)$ be an aggregation function obtained by fixing $\boldsymbol{b}_{-i}$ and $\boldsymbol{p}$, and let $T_+$ and $T_-$ be the sets of undersampled and oversampled tokens for agent $i$. Then we say that an implementation $\sigma$ is stable with respect to aggregation function $q$ if for any $r \in \mathcal{R}$ there are two tokens $u_r \in T_+$ and $o_r \in T_-$ and a threshold $\theta_r \in \mathbb{R}_+ \cup \{\infty\}$ such that:
$$\sigma(b_i, r) = o_r, \text{ if } b_i < \theta_r; \text{ and } \sigma(b_i, r) = u_r, \text{ if } b_i \geq \theta_r.$$

Here is an example of stable sampling for a single undersampled token $u$ and a single overampled token $o$. Sample $r$ uniformly in $[0, 1]$. If $q_o(b_i) < r$ assign $o$; assign $u$ otherwise. Note that this implementation is: (i) deterministic for fixed $r$; (ii) monotone in the bid; (iii) matches the probabilities of $q$ in expectation. The following theorem generalizes this to any number of tokens.

THEOREM 3.12. *Given a monotone distribution aggregation function $q$ then for any agent $i$ and fixed $\boldsymbol{b}_{-i}$ and $\boldsymbol{p}$ there always exists a stable implementation $\sigma$ of $q$.*

Proof. Given an aggregation $q(b_i)$ we construct a stable sampling procedure $\sigma$. Let $Q_+(b_i)$ and $Q_-(b_i)$ be the probability of sampling tokens from $T_+$ and $T_-$:

$$Q_+(b_i) = \sum_{t \in T_+} q_t(b_i), \qquad Q_-(b_i) = \sum_{t \in T_-} q_t(b_i).$$

Both are monotone as $q$ is monotone (by Lemma 3.10). Reparameterize functions $q_t(b_i)$ in the range $I = [Q_+(0), Q_+(\infty)]$ by defining:[4]

$$\hat{q}_t(x) = q_t(Q_+^{-1}(x)), \forall t \in T, x \in I.$$

Since $\hat{q}_t(x)$ is monotone, by Lebesgue's Differentiation Theorem, it is differentiable almost everywhere on $I$. We observe that:

$$\sum_{t \in T_+} \hat{q}_t(x) = Q_+(Q_+^{-1}(x)) = x, \ \sum_{t \in T_-} \hat{q}_t(x) = Q_-(Q_+^{-1}(x)) = 1 - x.$$

Then $\hat{q}_t'(x)$ for $t \in T_+$ forms a probability distribution over $T_+$. Similarly $-\hat{q}_t'(x)$ for $t \in T_-$ forms a probability distribution over $T_-$. Define $\kappa^+(x), \kappa^-(x) \in \Delta(T)$ as $\kappa_t^+(x) = \hat{q}_t'(x)$ for $t \in T_+$, and zero otherwise and $\kappa_t^-(x) = -\hat{q}_t'(x)$ for $t \in T_-$ and zero otherwise.

We also define the vector $q^+, q^- \in \Delta(T)$ such that $q_t^+ = q_t(0)/Q_+(0)$ for $t \in T_+$ and zero for $t \in T_-$. Similarly: $q_t^- = q_t(\infty)/Q_+(\infty)$ for $t \in T^-$ and zero otherwise.

Finally, we define a deterministic function

$$\text{Sampler} : \Delta(T) \times [0,1] \to T$$

that takes a probability vector $p \in \Delta(T)$ and $r \in [0,1]$ and outputs an index $t \in T$ such that $\sum_{j<t} p_j < r \le \sum_{j \le t} p_j$.

Now, we are ready to define the stable sampling procedure. Let $\mathcal{R}$ be the uniform distribution over $[0,1]^2$. Given $r = (r_A, r_B) \sim \mathcal{R}$ we define the output $t = \sigma(b_i, r)$ as follows:

(1) if $r_A \le Q_+(0)$, $t = \text{Sampler}(q^+, r_B) \in T_+$;
(2) if $Q_+(0) < r_A \le Q_+(b_i)$, $t = \text{Sampler}(\kappa^+(r_A), r_B) \in T_+$;
(3) if $Q_+(b_i) < r_A \le Q_+(\infty)$, $t = \text{Sampler}(\kappa^-(r_A), r_B) \in T_-$;
(4) if $Q_+(\infty) < r_A$, $t = \text{Sampler}(q^-, r_B) \in T_-$.

Since $\sigma(b_i, r)$ is deterministic, for any fixed $r$, either the output can not be influenced by the bid ($r_A < Q_+(0)$ or $r_A > Q_+(\infty)$) or it can only cause the output to shift from an oversampled token $\text{Sampler}(\kappa^-(r_A), r_B)$ to an undersampled token $\text{Sampler}(\kappa^+(r_A), r_B)$.

We now argue that tokens are sampled with the correct probabilities. For $t \in T_+$, the total probability of getting sampled is:

$$\int_0^{Q_+(0)} q_t^+ \mathrm{d}r_A + \int_{Q_+(0)}^{Q_+(b)} \hat{q}_t'(r_A) \mathrm{d}r_A = q_t(0) + \hat{q}_t(Q_+(b)) - \hat{q}_t(Q_+(0))$$

$$= q_t(0) + q_t(b) - q_t(0) = q_t(b).$$

Similarly for tokens in $T_-$, the probably of being sampled is:

$$\int_{Q_+(b)}^{Q_+(\infty)} -\hat{q}_t'(r_A) \mathrm{d}r_A + \int_{Q_+(\infty)}^1 q_t^- \mathrm{d}r_A$$
$$= \hat{q}_t(Q_+(b)) - \hat{q}_t(Q_+(\infty)) + q_t(\infty) = q_t(b) - q_t(\infty) + q_t(\infty) = q_t(b).$$

This completes the proof. □

*3.3.2 Second Price via Stable Sampling.* A stable implementation of a monotone aggregation rule suggests a natural pricing rule: For any given randomness $r$, if the oversampled token $o_r$ is sampled, the agent pays zero as it is the same token that would have been sampled if their bid was zero. If the agent's bid was high enough to move from the oversampled to the undersampled token $u_r$, then the agent pays the critical bid $\theta_r$.

---

[4]Here $Q_+^{-1}(x)$ refers to a generalized inverse (or the quantile function [7]) so that it is properly defined even when $Q_+(x)$ is discontinuous.

It has the property that the payment is proportional to the extent to which it shifts the distribution $q(b_i)$ towards the desired distribution $p_i$. Interestingly, the expected payment does not depend on the actual implementation chosen. Moreover, the expected payment corresponds to the standard Myersonian payment formula where the distribution is replaced by the total variation distance between the agent's preferred distribution $p_i$ and the implemented distribution $q(b_i)$. We again omit the terms $b_{-i}, p$ since they are fixed in each context. If a token $t \in T_-$ is sampled, the payment is naturally $\zeta_i(b_i, t) = 0$. For a token $t \in T_+$ we have:

$$\zeta_i(b_i, t) q_t(b_i) = \mathbb{E}_r [\theta_r \cdot \mathbf{1}\{\sigma(b_i, r) = t\}]$$
$$= \mathbb{E}_r \int_0^{b_i} \mathbf{1}\{\sigma(b_i, r) = t\} - \mathbf{1}\{\sigma(b', r) = t\} \mathrm{d}b' = \int_0^{b_i} q_t(b_i) - q_t(b') \mathrm{d}b'.$$

Hence:

$$z_i(b_i) = \sum_t \zeta_i(b_i, t) q_t(b_i) = \int_0^{b_i} \sum_{t \in T_+} q_t(b_i) - \sum_{t \in T_+} q_t(b') \mathrm{d}b'$$
$$= \frac{1}{2} \int_0^{b_i} \|q(b_i) - p_i\|_1 - \|q(b') - p_i\|_1 \mathrm{d}b'.$$

*Counterfactuals.* The practical advantage of a stable sampling implementation coupled with a second price rule is to offer advertisers a description where it is clear that they only pay if they can improve the outcome. Moreover, advertisers can more easily evaluate counterfactuals with a fixed $r$, where they can only produce one of two outcomes on each token that can easily be compared.

*Universally Stable Sampling.* We chose to define stable sampling as a single-agent algorithm with fixed $b_{-i}$. One may define a universal stable implementation as $\sigma^{\text{univ}} : \mathbb{R}^n \times \mathcal{R} \to T$ such that

$$\forall b \in \mathbb{R}_+^n, t \in T, \ \Pr_{r \sim R}[\sigma^{\text{univ}}(b, r) = t] = q_t,$$

and for any $i$ and $b_{-i}$, $\sigma^{\text{univ}}(\cdot, b_{-i}, r)$ is stable. In Appendix D, we provide counter-examples where such $\sigma^{\text{univ}}$ do not always exist.

# 4 DESIGN OF AGGREGATION FUNCTIONS

In the previous section, we discussed payment schemes and incentive properties assuming we have an aggregation function. Here we investigate the design of aggregation functions. We adopt two guiding principles: (1) We first define a welfare function to model the overall satisfaction of the agents with the final distribution $q$, weighted by their bids $b_i$. The welfare function has the form:

$$\text{Wel}(p, b, q) = \sum_i b_i \rho(p_i, q),$$

where $\rho : \Delta(T) \times \Delta(T) \to \mathbb{R}$ indicates how close the distribution $q$ is to the preferred $p_i$. (2) The second is to define the closeness function $\rho$ based on the typical loss functions in LLM training.

## 4.1 Review of LLM training

So far we assume that an LLM $f : T^* \to \Delta(T)$ is already trained. In order to discuss the training process, it is useful to recall that an LLM is a neural network parameterized by a vector of weights $W \in \mathbb{R}^N$ in a very high dimensional space. To discuss training, it is useful to think of $f$ as a function of both input and weights:

$$f : T^* \times \mathbb{R}^N \to \Delta(T).$$

We represent the second argument by a superscript $W$. Training is to optimize $W$ such that $f^W(\cdot)$ minimizes a certain loss function over a dataset. A dataset is a sequence of pairs $(x_i, y_i)$ with $x_i \in T^*$

(input sequence) and $y_i \in T$ (label), and a loss function is a function $\ell : T \times \Delta(T) \to \mathbb{R}$. A network typically seeks to minimize:

$$\min_W \sum_i \ell(y_i, f^W(x_i)).$$

A widely used loss function in the first stage is the KL-divergence:

$$\ell_{\mathsf{KL}}(y, x) = -\ln[f^W(y|x)]$$

where we use the notation $f(y|x)$ to represent the probabiliy that a token $y \in T$ is sampled from $f(x) \in \Delta(T)$.

It is useful to think of this problem in the limit where the size of the dataset grows to infinity and it can be effectively thought of as a full-support distribution over $T^* \times T$. In that setting, we can represent the dataset as a distribution $\mu \in \Delta(T^* \times T)$ over input-label pairs $(x, y) \in T^* \times T$. We will write $\mu(x) = \sum_y \mu(x, y)$ to denote the marginal distribution on $x$ and $\mu(\cdot|x)$ to denote the conditional distribution of labels given an input $x$. Then

$$\min_W \mathcal{L}_{\mathsf{KL}}^\mu(f^W), \; \mathcal{L}_{\mathsf{KL}}^\mu(f) := \sum_{x \in T^*} \mu(x) \cdot D_{\mathsf{KL}}(\mu(\cdot|x)\|f(x)), \quad \text{(KL)}$$

where $D_{\mathsf{KL}}(p\|q) = \sum_t p(t) \ln \frac{p(t)}{q(t)}$ is the KL-divergence.

LLMs are typically trained through a successive refinement of weights: $W^{\mathsf{pre}} \to W^{\mathsf{SFT}} \to W^{\mathsf{RL}}$. In pre-training we compute $W^{\mathsf{pre}}$ by solving problem (KL) on a generic dataset $\mu^{\mathsf{pre}}$ via stochastic gradient descent. In the second stage, we initialize the weights as $W = W^{\mathsf{pre}}$ and run stochastic gradient descent to solve the same problem (KL) on a more specialized dataset $\mu^{\mathsf{SFT}}$. In other words, we solve the same problem on two different datasets.

The problem in the RLHF stage is different. The dataset only contains $x$ (still represented by $\mu$) and for any $y$, we have a function $r(x, y)$ giving the reward of mapping $x$ to $y$. Then $W^{\mathsf{RL}}$ is obtained by maximizing reward while minimizing the distance to the function $f^{\mathsf{SFT}}$ from previous stages (the PPO algorithm [12, 16]):

$$\max_W \mathcal{L}_{\mathsf{RL}}^{\mu,r}(f^W), \quad \text{(RL)}$$

$$\mathcal{L}_{\mathsf{RL}}^{\mu,r}(f) := \sum_{x \in T^*} \mu(x) \left[ \sum_y r(x, y) f(y|x) - \beta D_{\mathsf{KL}}(f(x)\|f^{\mathsf{SFT}}(x)) \right].$$

*KL-divergence and entropy.* For the next propositions it is useful to recall that the entropy of a distribution $p \in \Delta(T)$ is $H(p) = -\sum_{t \in T} p(t) \ln p(t)$. Given two distributions $p, q \in \Delta(T)$, the cross entropy of $q$ relative to $p$ is $H(p, q) = -\sum_{t \in T} p(t) \ln q(t)$. Hence we can write $D_{\mathsf{KL}}(p\|q) = H(p, q) - H(p)$. We will also use Gibbs' inequality $H(p) \leq H(p, q), \forall p, q \in \Delta(T)$.

## 4.2 KL-inspired aggregation

The first aggregation method will be based on the (KL) program. When trying to aggregate LLMs $f_i$ according to bids $b_i$, we will design a function $q$ that mimics the outcome of the following thought experiment. We will imagine that each LLM was obtained by solving the (KL) on a dataset represented by $\mu_i$, where the marginal over inputs are the same $\mu_i(x) = \mu(x), \forall i$ but potentially differ on the marginals on the labels $\mu_i(y|x)$. In this thought experiment, we will combine their LLMs by re-training an LLM on the combined labels weighted by the bids. In other words, we will solve the problem:

$$\min_W \mathcal{L}_{\mathsf{KL}}^{\bar{\mu}}(f^W), \qquad \text{where } \bar{\mu} = \sum_i b_i \mu_i / \sum_i b_i. \quad \text{(3)}$$

The next proposition morally says that we can obtain a solution to the (KL) problem on the aggregated dataset (3) by combining its solutions on individual datasets. The proof appears in Appendix E.

PROPOSITION 4.1. *Consider datasets $\mu_i$ such that $\mu_i(x) = \mu(x), \forall i, x$ and let $\bar{\mu}$ be their weighted average. Let $f_i$ be the minimizer of $\mathcal{L}_{KL}^{\mu_i}$ and $f^*$ be the minimizer of $\mathcal{L}_{KL}^{\bar{\mu}}$, the loss on the aggregated dataset. Then $f^*$ is the solution to:*

$$\min_f \sum_x \mu(x) \sum_i b_i D_{\mathsf{KL}}(f_i(x)\|f(x)). \quad \text{(4)}$$

Proposition 4.1 motivates the following welfare function:

$$\mathrm{WEL}_{\mathsf{KL}} = \sum_{i \in [n]} b_i \cdot \rho_{\mathsf{KL}}(p_i, q) = -\sum_{i \in [n]} \sum_{t \in T} b_i \cdot p_i(t) \cdot \ln \frac{p_i(t)}{q(t)}.$$

Now, we characterize the aggregation function that optimizes $\mathrm{WEL}_{\mathsf{KL}}$. The proof in Appendix E uses Gibb's inequality.

LEMMA 4.2. *The efficient aggregation function that maximizes $\mathrm{WEL}_{\mathsf{KL}}$ is the linear combination of $p_i$:*

$$\forall t \in T, \; q_{\mathsf{KL}}(t) = \sum_i b_i \cdot p_i(t) / \sum_i b_i.$$

Besides being monotone and aligned with how LLMs are trained, this aggregation function has the advantage that in order to sample a token from it, we only need to call a single LLM. We can choose an index $i$ proportionally to the bids and then sample a token from the $i$-th LLM. To compute second price payments, however, we still need to query the LLMs for all agents.

## 4.3 RL-inspired aggregation

Now consider a different thought experiment where all agents use the same pre-trained and fine-tuned model of weights $W^{\mathsf{SFT}}$ but each one uses a different reward function $r_i(x, y)$ for RLHF. We combine their LLMs by solving the (RL) problem on the combined reward functions weighted by the bids. We will solve the problem:

$$\max_W \mathcal{L}_{\mathsf{RL}}^{\mu,\bar{r}}(f^W), \quad \bar{r} = \sum_i b_i r_i / \sum_i b_i.$$

Analogously to Proposition 4.1, we show that we can obtain the solution to the aggregated problem by combining the solutions on each dataset. We defer the proof of Proposition 4.3 to Appendix E.

PROPOSITION 4.3. *Consider datasets $\mu, r_i$ and let $f^{\mathsf{SFT}}$ be the solution of program (KL) with data $\mu$. If $f_i$ is the maximizer of $\mathcal{L}_{RL}^{\mu,r_i}$, let $f^*$ be the maximizer of $\mathcal{L}_{RL}^{\mu,\bar{r}}$ where $\bar{r}$ is the weighted average of the reward functions, then $f^*$ is the function minimizing:*

$$\min_f \sum_x \mu(x) \sum_i b_i D_{\mathsf{KL}}(f(x)\|f_i(x)). \quad \text{(5)}$$

Similar to what we did in the previous subsection, we can also define a welfare function inspired by Proposition 4.3. Namely:

$$\overline{\mathrm{WEL}}_{\mathsf{KL}} = \sum_{i \in [n]} b_i \cdot \rho_{\mathsf{KL}}(q, p_i) = -\sum_{i \in [n]} \sum_{t \in T} b_i \cdot q(t) \cdot \ln \frac{q(t)}{p_i(t)}.$$

LEMMA 4.4. *The efficient aggregation function that maximizes $\overline{\mathrm{WEL}}_{\mathsf{KL}}$ is the log-linear combination of $p_i$:*

$$\forall t \in T, \; \ln \bar{q}_{\mathsf{KL}}(t) = C + \frac{\sum_i b_i \ln \cdot p_i(t)}{\sum_i b_i},$$

*where $C$ is a normalization constant such that $\sum_t \bar{q}_{\mathsf{KL}}(t) = 1$.*

The proof follows directly from the proof of Proposition 4.3.

While not monotone (Example 3.9), the log-linear aggregation function is a reasonable choice assuming that the agents' preferences are aligned with the KL-divergence loss for the RL-stage training.

| $\lambda$ | Linear aggregation function $q_{KL}$ | Log-linear aggregation function $\bar{q}_{KL}$ |
|---|---|---|
| 1 | **Alpha Airlines**: Your ticket to paradise. | |
| 0.75 | **Alpha Airlines**: Fly to Hawaii and experience the beauty of the islands with aloha. **Beta Resorts**: Stay at our resorts and enjoy the best of Hawaii. | Feel the magic of Hawaii with a flight on **Alpha Airlines**, now offering 20% off all flights when you book with us today! |
| 0.6 | **Alpha Airlines** flies you to Hawaii, where you can enjoy a week-long stay at the **Beta resort** for just $1000. | Experience the magic of Hawaii with a flight on **Alpha Airlines**, now offering 20% off all flights when you book with us today! |
| 0.55 | **Alpha Airlines** flies you to Hawaii, where you can enjoy a week-long stay at the **Beta resort**. | Escape to the tropical paradise of Hawaii with [**Alpha Airlines**]! |
| 0.5 | **Alpha Airlines** flies you to Hawaii, where you can enjoy a beautiful sunset on the beach. Stay 3 nights and get the 4th free at the **Beta Resort**. | Experience the magic of Hawaii with a stay at the luxurious [**Beta Resort**] and a refreshing flight on [**Alpha Airlines**]. |
| 0.45 | Fly **Alpha Airlines** to sunny Hawaii and enjoy the secluded beaches and private lagoons of the Royal Hawaiian **Beta Resort**. | Experience the magic of Hawaii with a stay at the luxurious [**Beta Resort**] and a special flight offer from [**Alpha Airlines**]. |
| 0.4 | Fly **Alpha Airlines** to sunny Hawaii and enjoy the first-class treatment that awaits you at **Beta Resort**, all for one low price. | Experience the magic of Hawaii at the [**Beta Resort**], where you'll feel like you're in a tropical paradise. |
| 0.25 | Experience the magic of Hawaii at the **Beta Resort**, where the sun shines brighter and the waves crash louder — book your stay today with our exclusive 20% off discount! | Experience the magic of Hawaii at the **Beta Resort**, where you'll be pampered like royalty and surrounded by breathtaking beauty. |
| 0 | Hawaii's **Beta Resort**: a paradise where the sun shines brighter, the waves sing sweeter, and the sand feels softer. | |

**Table 2: Text generation from two aggregation functions with different $\lambda = b_1/(b_1 + b_2)$.**

## 5 DEMONSTRATION

We implement the aggregation functions proposed in Section 4 and discuss the examples they produce. Off-the-shelf LLMs respond full text passages. In our case, we need to peak at the internal states of LLMs (the probability distributions over tokens) at each token generation stage. For that reason, we use a custom version of the [REDACTED FOR BLIND REVIEW] model with a custom inference method that allows us to access the distributions. The aggregation functions are implemented inside the inference method.

Starting from the same base model, we customize it for different agents by prompt-tuning. We start with a based model $f : T^* \to \Delta(T)$ and for each agent $i$ we come up with a "prompt" $s_0^i \in T^*$ and now we define for each agent $i$ the LLM $f_i : T^* \to \Delta(T)$ as:

$$f_i(s) = f(s_0^i \oplus s)$$

Therefore if $\tau_1, \ldots, \tau_{k-1}$ are the first $k - 1$ tokens generated, then the preferred distribution of agent $i$ over the $k$-th token is given by:

$$p_i = f(s_0^i \oplus s \oplus \tau_1 \oplus \cdots \oplus \tau_{k-1}),$$

One key advantage of simulating LLM agents with different prompts is that one does not need to serve multiple LLMs at the same time, but instead making multiple queries to the served one. Because of their large sizes, serving multiple LLMs can be very costly and practically challenging. As one of the key strengths of LLMs, the flexibility to accomplish various tasks with properly designed prompts sheds light to the possibility of training one universal LLM that can, for example, generate different ads according to agent-specific prompts. That is, the universal advertising LLM, plus an advertiser-specific prompt, behaves like an advertiser-specific LLM through the online in-context few-shot learning.

## 5.1 Setup

The example we show here involves two agents, each of them would like to advertise for their brands, "Alpha Airlines" and "Beta Resort",

regarding a shared topic "Hawaii". We choose fictitious brands to avoid the model directly using any existing ads. We use the brand names "Alpha" and "Beta" that do not have strong meanings on their own to minimize any potential hallucination, as we are using a common purposed LLM that is not optimized for our task. Each agent is given the following prompt:

> "You are an expert of writing texts that naturally combines two ads together. Your choice of words and sentences is full of artistic flair.
> Write a one-sentence ad for ______."

Agent A uses "a flight to Hawaii using [**Alpha Airlines**]" to fill the blank, while agent B uses "a vacation in Hawaii at the [**Beta Resort**]". The first two sentences in the prompt aim to improve the quality of the ad generation through *assigning roles* (see, for example, [24]).

*Two bids as one parameter.* Since in both the linear aggregation rule $q_{KL}$ and the log-linear aggregation rule $\bar{q}_{KL}$, there is only one degree of freedom, we parameterize the response by $\lambda = b_1/(b_1+b_2)$.

## 5.2 Results

The results are listed in Table 2, where from top to bottom, the value of $\lambda$ decreases from 1 to 0. As we can see for both aggregation functions, the generated texts roughly follow the pattern of "only Alpha Airlines" → "both Alpha Airlines and Beta Resort" → "only Beta Resort" when $\lambda$ goes from 1 to 0. This is expected, as $\lambda$ going from 1 to 0 corresponds to $b_2$ increasing from 0 to $\infty$ with $b_1$ fixed (or $b_1$ decreasing from $\infty$ to 0 with $b_2$ fixed). The thresholds of pattern changes are 0.75 and 0.4 for the linear aggregation, and 0.5 and 0.45 for the log-linear aggregation.

We emphasize that the examples are generated with a general purposed LLM, and it is reasonable to believe that the performance can be improved with proper fine-tuning for specific tasks.

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

## A   ADDITIONAL RELATED WORK

To the best of our knowledge, the exact research question and our approaches in this work have not been studied previously. However, our work is indeed connected to a few lines of research. The most relevant to us is perhaps the recent growing literature on fine-tuning LLMs, with the reinforcement learning from human feedback (RLHF) as a representative approach [2, 3, 12, 21]. At a high level, fine-tuning and RLHF seek to align a generally pre-trained LLM with certain desirable behaviors. This is in spirit analogous to our goal of designing LLMs to better align with a group of agents' overall preferences. However, our research challenges and methods are both different from fine-tuning. Specifically, fine-tuning refines the underlying model's parameters whereas our approach is one-layer up and directly aggregates the output distributions from multiple models. The main challenge we address is the potential incentive misalignment while eliciting LLM agents' preferences, whereas human labelers or other models that generate reward feedback for RLHF are assumed to be genuine and do not misrepresent their own preferences. In-context learning [6, 22, 23] is similar to us in the sense that they also do not change the model parameters. They influence token distributions by conditioning on better-generated prefix contexts, whereas our approach directly aggregates distributions from multiple LLM agents.

Another related line of research is the celebrated field of mechanism design (MD), particularly for the choice of a "public project" [9, 13] (which is the output of the designed LLM in our situation) that maximizes a certain welfare function. Similar to these type of design problem, a core challenge in our problem is to elicit truthful preferences from unknown agents. However, the design problem in our case is fundamentally different — we choose a high-dimensional distribution from an $\mathbb{R}^T$ space with only partial knowledge about agents' preferences whereas previous MD for public project typically pick a choice from a discrete (often exponentially large) set with clear agent valuation functions [9, 13]. From this perspective, our work also bear some similarity to the rich literature of robust mechanism design. Most of robust MD literature still assume existence of value functions with uncertainty modeled by Bayesian beliefs or in a max-min sense [4, 5, 8, 15]. However, assuming such a valuation function over tokens or their distributions does not appear realistic in creatives generation, thus our model is more similar to a worst-case style consideration during which we only assume partial yet "obvious" preferences.

## B   OMITTED PROOFS FROM SECTION 3.2

### Proof of Lemma 3.6

PROOF. Since the distribution aggregation function $q$ is consistent, we can define a partial order $\succeq_{i,\boldsymbol{p}}$ over $\mathbb{R}_+$ implied by the order over $Q(\boldsymbol{b}_{-i}, \boldsymbol{p})$ (assumed by the lemma) such that,

$$\forall b_i, b_i' \in \mathbb{R}_+, \ b_i \succeq_{i,\boldsymbol{p}} b_i' \iff q(b_i, \boldsymbol{b}_{-i}, \boldsymbol{p}) \succeq_i q(b_i', \boldsymbol{b}_{-i}, \boldsymbol{p}).$$

By assumption of the lemma, $\succeq_i$ establishes a total order over $Q(\boldsymbol{b}_{-i}, \boldsymbol{p})$. We argue that this implies that $\succeq_{i,\boldsymbol{p}}$ will be a total order over $\mathbb{R}_+$. Concretely, for every pair $b_i, b_i' \in \mathbb{R}_+$ we have $q(b_i), q(b_i') \in Q(\boldsymbol{b}_{-i}, \boldsymbol{p})$. So either $q(b_i) \succeq_i q(b_i')$ or $q(b_i') \succeq_i q(b_i)$, due to the lemma's assumption of total order over $Q(\boldsymbol{b}_{-i}, \boldsymbol{p})$. Hence every pair

$b_i, b_i' \in \mathbb{R}_+$ has an order under $\succeq_{i,p}$, so it must be a total order over $\mathbb{R}_+$.

Consequently, there exists a bijection $\langle f_{i,p}, f_{i,p}^{-1} \rangle$ between $\mathbb{R}_+$ and $\mathbb{R}_+$ such that,

$$\forall b_i, b_i' \in \mathbb{R}_+, \ b_i \succeq_{i,p} b_i' \iff f_{i,p}(b_i) \geq f_{i,p}(b_i').$$

Letting $\pi_i(b) = f_{i,p}^{-1}(b)$, we have that $\tilde{q}(b_i, b_{-i}, p) = q(\pi_i(b_i), b_{-i}, p)$ is a monotone distribution aggregation function for agent $i$. Applying the same argument and relabeling procedure for every $i$ from 1 to $n$, we completed the proof. □

## Proof of Lemma 3.7

PROOF. We first prove the "only if" ("$\Longrightarrow$") direction. That is, suppose $\mathcal{M} = \langle q, z \rangle$ is payment-monotone, then it must imply a total order over $Q(b_{-i}, p) = \{q(b_i, b_{-i}, p) : b_i \in \mathbb{R}\}$ for any fixed $b_{-i}$ and $p$.

Fix any $b_{-i}$ and $p$. For any $q, q' \in Q(b_{-i}, p) = \{q(b_i, b_{-i}, p) : b_i \in \mathbb{R}\}$ such that $q = q(b_i, b_{-i}, p)$, $q' = q(b_i', b_{-i}, p)$. Let $z_i = z_i(b_i, b_{-i}, p)$ and $z_i' = z_i(b_i', b_{-i}, p)$ be the corresponding payment given by $\mathcal{M}$. Without loss of generality, suppose $z_i \geq z_i'$. Then by payment-monotonicity of the mechanism $\mathcal{M}$, we have $q \succeq_i q'$. In other words, $\succeq_i$ establishes a total order over $Q$.

Next we show the "if" ("$\Longleftarrow$") direction. That is, given a total order over $Q(b_{-i}, p) = \{q(b_i, b_{-i}, p) : b_i \in \mathbb{R}\}$ for any fixed $b_{-i}$ and $p$, we can construct a payment rule $z$ such that $\mathcal{M} = \langle q, z \rangle$ is payment-monotone.

Fix any $b_{-i}$ and $p$. For any $q, q' \in Q(b_{-i}, p)$, since $\succeq_i$ establishes a total order over $Q(b_{-i}, p)$, we must have either $q \succeq_i q'$ or $q' \succeq_i q$. Without loss of generality, suppose $q \succeq_i q'$. Since $\succeq_i$ establishes a total order over $Q(b_{-i}, p)$, which is isomorphic to a subset of $\mathbb{R}_+$ by definition of $Q$, this implies the existence of a bijection $f_{i,b_{-i},p}$ between $Q(b_{-i}, p)$ and some subset of $\mathbb{R}$ such that $\forall q, q' \in Q(b_{-i}, p)$,

$$q \succeq_i q' \iff f_{i,b_{-i},p}(q) \geq f_{i,b_{-i},p}(q'),$$

where "$\geq$" here is in the standard order of real numbers. Letting $z_i(b, p) = f_{i,b_{-i},p}(q(b, p))$, we obtain a payment-monotone mechanism $\langle q, z \rangle$. □

## Proof of Theorem 3.5

PROOF. Applying Lemma 3.7, we know that the payment monotonicity of $\mathcal{M}$ implies a total order over $Q(b_{-i}, p)$. Then applying Lemma 3.6, we know that there exists a bijection $\pi_i : \mathbb{R}_+ \to \mathbb{R}_+$ for each $i$ such that $\tilde{q}$ — defined via $\tilde{q}(b, p) = q(\pi(b), p)$ for $b \in \mathbb{R}_+^n$ and $p \in \Delta(T)^n$ — is a monotone aggregation function.

Now let us further define $\tilde{z}(b, p) = z(\pi(b), p)$ for each $b \in R_+^n$ and $p \in \Delta(T)^n$. Since $\pi$ is a bijection, $\mathcal{M} = \langle q, z \rangle$ and $\tilde{\mathcal{M}} = \langle \tilde{q}, \tilde{z} \rangle$ are strategically equivalent by definition.

It remains to show that the mechanism $\tilde{\mathcal{M}} = \langle \tilde{q}, \tilde{z} \rangle$ is payment-monotone. We know that the orignial mechanism $\mathcal{M} = \langle q, z \rangle$ satisfies payment monotonicity, meaning that for each $p$, $b_{-i}$, and $b_i \geq b_i'$,

$$z_i(b_i, b_{-i}, p) \geq z_i(b_i', b_{-i}, p) \iff q(b_i, b_{-i}, p) \succeq_i q(b_i', b_{-i}, p).$$

But then, with $b = (b_i, b_{-i})$ and $b' = (b_i', b_{-i})$, we also have

$$\tilde{z}_i(b, p) = z_i(\pi(b), p) \geq z_i(\pi(b')) = \tilde{z}_i(b', p)$$
$$\iff \tilde{q}(b, p) = q(\pi(b), p) \succeq_i q(\pi(b'), p) = \tilde{q}(b', p),$$

so the pair $\tilde{q}, \tilde{z}$ satisfies payment monotonicity as needed. □

## C OMITTED PROOFS FROM SECTION 3.3

### Proof of Lemma 3.10

PROOF. We first prove the "only if" ("$\Longrightarrow$") direction. Suppose $q$ is a monotone distribution aggregation function. By Definition 3.3, for any agent $i$ and $b_i' \geq b_i \geq 0$, we have

$$q(b_i', b_{-i}, p) \succeq_i q(b_i, b_{-i}, p) \succeq_i q(0, b_{-i}, p).$$

For any undersampled token $t \in T_+$, because $q_t(0, b_{-i}, p) \leq (p_i)_t$, then by Definition 2.1, we have

$$q_t(b_i, b_{-i}, p), q_t(b_i', b_{-i}, p) \in [q_t(0, b_{-i}, p), (p_i)_t].$$

Hence by $q(b_i', b_{-i}, p) \succeq_i q(b_i, b_{-i}, p)$, we have

$$q_t(b_i', b_{-i}, p) \geq q_t(b_i, b_{-i}, p),$$

namely, $q_t(b_i, b_{-i}, p)$ weakly increases with $b_i$ and never goes above $(p_i)_t$.

Similarly, we can prove that for any oversampled token $t \in T_-$, $q_t(b_i, b_{-i}, p)$ weakly decreases with $b_i$ and never goes below $(p_i)_t$.

Then we prove the "if" ("$\Longleftarrow$") direction. Consider any $b_i' \geq b_i$. For any undersampled token $t \in T_+$, as $q_t(b_i, b_{-i}, p) \leq (p_i)_t$ weakly increases with $b_i$, we have

$$q_t(b_i, b_{-i}, p) \leq q_t(b_i', b_{-i}, p) \leq (p_i)_t.$$

Similarly, we have for any oversampled token $t \in T_-$,

$$q_t(b_i, b_{-i}, p) \geq q_t(b_i', b_{-i}, p) \geq (p_i)_t.$$

Then by Definition 2.1,

$$q(b_i', b_{-i}, p) \succeq_i q(b_i, b_{-i}, p),$$

which then implies the monotonicity of $q$. □

## D UNIVERSALLY STABLE SAMPLING

*Example D.1 (Counterexample 4-token).* Consider two agents $\{1, 2\}$ and 4 tokens $\{t_1, t_2, t_3, t_4\}$. Assume that both agents have the same preferred distribution $p_1 = p_2 = (0, 0, .5, .5)$ and the allocation function is such that if both agents bid zero the allocation is $(.5, .5, 0, 0)$. Hence both both agents have the same set of favored tokens $T_+ = \{t_3, t_4\}$ and less favored tokens $T_- = \{t_1, t_2\}$. The aggregation function $q(b_1, b_2, p_1, p_2)$ is given by the following table:

|          | $b_1 = 0$              | $b_1 = 1$              |
| -------- | --------------------- | --------------------- |
| $b_2 = 0$ | $q_{00} = (.5, .5, 0, 0)$ | $q_{10} = (0, .5, .5, 0)$ |
| $b_2 = 1$ | $q_{01} = (.5, 0, .5, 0)$ | $q_{11} = (0, 0, .5, .5)$ |

One can verify that the aggregation function is monotone: When either of the agents increase the bid from 0 to 1, exactly $1/2$ of the probability mass moves from $T_- = \{t_1, t_2\}$ to $T_+ = \{t_3, t_4\}$.

Now we show that there does not exist a universally stable sampling algorithm that implements this aggregation function. Suppose there exist one, $\sigma$, let $r_A = \{r | \sigma(q_{00}, r) = t_1\}$ and $r_B = \{r | \sigma(q_{00}, r) = t_2\}$. Because $\sigma$ is stable for bidder 1, when bidder 1 increases bid, the probability mass only transfers from $T_-$ to $T_+$.

In this case, when the bid profile $(b_1, b_2)$ moves from $(0, 0)$ to $(1, 0)$, we must have

$$\sigma(q_{10}, r) = \begin{cases} t_3, & r \in r_A \\ t_2, & r \in r_B \end{cases}.$$

Further apply the same argument when $(b_1, b_2)$ moves from $(1, 0)$ to $(1, 1)$, we have

$$\sigma(q_{11}, r) = \begin{cases} t_3, & r \in r_A \\ t_4, & r \in r_B \end{cases}.$$

However, if we consider $(b_1, b_2)$ moves along the path $(0, 0) \rightarrow (0, 1) \rightarrow (1, 1)$, we should have

$$\sigma(q_{01}, r) = \begin{cases} t_1, & r \in r_A \\ t_3, & r \in r_B \end{cases}, \qquad \sigma(q_{11}, r) = \begin{cases} t_4, & r \in r_A \\ t_3, & r \in r_B \end{cases}.$$

We end up with a contradiction on the value of $\sigma(q_{11}, r)$ while moving from $(0, 0)$ to $(1, 1)$ along two different paths.

*Example D.2 (Counterexample 3-token).* Consider two agents $\{1, 2\}$ and 3 tokens $\{t_1, t_2, t_3\}$, where the agents have different sets of favored (less favored) tokens. In particular, $T_1^+ = \{t_1, t_3\}$, $T_1^- = \{t_2\}$ and $T_2^+ = \{t_3\}$, $T_2^- = \{t_1, t_2\}$. The aggregation function is given by the following table:

|  | $b_1 = 0$ | $b_1 = 1$ |
|---|---|---|
| $b_2 = 0$ | $q_{00} = (.5, .5, 0)$ | $q_{10} = (.5, 0, .5)$ |
| $b_2 = 1$ | $q_{01} = (0, .5, .5)$ | $q_{11} = (.5, 0, .5)$ |

One can verify that the aggregation function is monotone: When $b_1$ increases from 0 to 1, exactly $1/2$ of the probability mass moves from $t_2$ to $t_3$ (when $b_2 = 0$) or $t_1$ (when $b_2 = 1$). When $b_2$ increases from 0 to 1, either $1/2$ of the probability mass moves from $t_1$ to $t_3$ (when $b_1 = 0$) or no move (when $b_1 = 1$).

Similarly, suppose that there exists a universally stable sampling algorithm $\sigma$ that implements $q$. Let $r_A = \{r | \sigma(q_{00}, r) = t_1\}$ and $r_B = \{r | \sigma(q_{00}, r) = t_2\}$.

Following the same argument in Example D.1, consider the bid profile $(b_1, b_2)$ moves along the path $(0, 0) \rightarrow (1, 0) \rightarrow (1, 1)$, we must have

$$\sigma(q_{10}, r) = \sigma(q_{11}, r) = \begin{cases} t_1, & r \in r_A \\ t_3, & r \in r_B \end{cases}.$$

However, consider the bid profile $(b_1, b_2)$ moves along the path $(0, 0) \rightarrow (1, 0) \rightarrow (1, 1)$, we must have

$$\sigma(q_{01}, r) = \begin{cases} t_3, & r \in r_A \\ t_2, & r \in r_B \end{cases}, \qquad \sigma(q_{11}, r) = \begin{cases} t_3, & r \in r_A \\ t_1, & r \in r_B \end{cases}.$$

We end up with a contradiction on the value of $\sigma(q_{11}, r)$.

## E  OMITTED PROOFS FROM SECTION 4

### Proof of Proposition 4.1

PROOF OF PROPOSITION 4.1. We prove the theorem by showing that the loss $\mathcal{L}_{KL}^{\bar{\mu}}$ and the loss in equation (4) differ by a constant and hence have the same minimizer. For $B = \sum_i b_i$ and a fixed $x$ we will show that:

$$B \cdot D_{KL}(\sum_i \tfrac{b_i}{B} \mu_i(\cdot | x) \| f^W(x)) - \sum_i b_i D_{KL}(f_i(x) \| f^W(x)) = \text{const}.$$

For notation simplicity, we omit the parameters $x$ and $W$ when it is clear from the context and write $\sum_i b_i f_i(x)/B = \bar{f}(x)$. Below we treat any term that doesn't depend on $f$ as a constant:

$$\sum_i b_i D_{KL}(f_i \| f) = \sum_i b_i H(f_i) - \sum_y b_i f_i(y|x) \ln f(y|x)$$

$$= -B \sum_y \frac{\sum_i b_i f_i(y|x)}{B} \cdot \ln f(y|x) + \sum_i b_i H(f_i)$$

$$= -B \cdot H(\bar{f}, f) + \sum_i b_i H(f_i)$$

$$= B \cdot D_{KL}(\bar{f} \| f) - B \cdot H(\bar{f}) + \sum_i b_i H(f_i)$$

$$= B \cdot D_{KL}(\bar{f} \| f) - \text{const}.$$

To complete the proof, observe that if $f_i$ is the unconstrained minimizer of $\mathcal{L}_{KL}^{\mu_i}(f)$ we must have $f_i(y|x) = \mu_i(y|x)$. □

### Proof of Lemma 4.2

PROOF OF LEMMA 4.2. Let $B = \sum_i b_i$ and consider $W_{KL}$:

$$W_{KL} = - \sum_{i \in [n]} b_i \cdot (H(p_i, q) - H(p_i))$$

$$= \sum_{i \in [n]} b_i \cdot H(p_i) + \sum_{i \in [n]} b_i \cdot \sum_{t \in [T]} p_i(t) \ln q(t)$$

$$= \sum_{i \in [n]} b_i \cdot H(p_i) + \sum_{t \in [T]} \ln q(t) \sum_{i \in [n]} b_i \cdot p_i(t)$$

$$= \sum_{i \in [n]} b_i \cdot H(p_i) + B \cdot \sum_{t \in [T]} \frac{\sum_{i \in [n]} b_i \cdot p_i(t)}{B} \ln q(t)$$

$$= \sum_{i \in [n]} b_i \cdot H(p_i) - B \cdot H(q_{KL}, q)$$

By Gibbs' inequality, the cross entropy $H(q_{KL}, q)$ is minimized if and only if $q = q_{KL}$. Hence this is also the maximizer of $W_{KL}$. □

### Proof of Proposition 4.3

PROOF. For a fixed $x$, $f^*(y|x)$ can be obtained by solving:

$$\max_{f(\cdot | x)} \sum_y f(y|x) \bar{r}(x, y) - \beta D_{KL}(f(x) \| f^{SFT}(x))$$

$$\text{s.t.} \sum_y f(y|x) = 1 \text{ and } f(y|x) \geq 0.$$

By the standard KKT conditions, the solution has the form:

$$f^*(y|x) = f^{SFT}(y|x) e^{\bar{r}(x,y)/\beta} C_x.$$

where $C_x$ is a normalization constant to ensure $\sum_y f^*(y|x) = 1$. For the same reason, we have that:

$$f_i(y|x) = f^{SFT}(y|x) e^{r_i(x,y)/\beta} C_{i,x}.$$

If $f^\circ$ is the function minimizing problem (5), then we can apply KKT conditions to obtain:

$$f^\circ(y|x) = \exp\left( \frac{1}{B} \sum_i b_i \ln f_i(y|x) \right) \cdot C_x'$$

for normalization constants $C'_x$ and $B = \sum_i b_i$. Replacing the formula for $f_i(x)$ from the previous line, we obtain that:

$$f^\circ(y|x) = \exp\left(\frac{1}{B}\sum_i b_i\left(r_i(x,y)/\beta + \ln f^{\text{SFT}}(y|x)\right)\right) \cdot C''_x$$

$$= \exp\left(\bar{r}(x,y)/\beta + \ln f^{\text{SFT}}(y|x)\right) \cdot C''_x = f^*(y|x).$$

This completes the proof. □

