# OpenReview forum: "Mechanism Design for Large Language Models"
_ACM.org/TheWebConf/2024/Conference — TheWebConf24 Oral_

### Official Review · Reviewer_KiYJ · 2023-11-20

**Novelty:** 6
**Technical Quality:** 5

**Review:**

The paper studies a scenario of online ad auctions when the bidding agents are LLMs. They propose a formalism for studying these auctions and propose auctions that can merge content from different advertises, thus effectively allowing for more bidders to win the auction.
The paper technically focuses on the domain of text generation, but it seems (as also noted in footnote 2) that the abstraction may be useful for other domains as well.
I believe the paper touches an interesting and certainly relevant topic of automated agents (and in particular, LLMs) operating in economic online markets and makes an interesting proposal for how to auction ad position to such automated agents. I have some questions about the assumptions and economic modeling in this paper as well as some other comments (see Questions below). The paper does make progress on this important topic of markets with automated agents and makes an interesting and novel contribution as to how to theoretically model LLM agents. The theoretical analysis seems to be well performed, given the assumptions, as far as I saw. Overall, I think the paper is interesting and believe that some of the economic issues that arise may be addressed by follow-up work. I would be happy to hear the authors view on the points raised in "Questions" below.

Minor comments:

- The way I understand it, the main motivation for not using a standard auction design is that there is an opportunity to improve welfare by having more than one winner even when there is a single item for sale (a single ad position). I think it would be good to emphasize further this motivation of improving welfare. Currently in paragraph 2 it is not very clear why the simple auction is not a good-enough solution.
- In the first time the acronym LLM appears, it would be better to present it.
- Line 190, the statement is not very clear. Did the authors mean the following? If for two different bids x and y of the same agent, the final distribution when the bid is x is closer to the preferred distribution than the final distribution when the bid is y for some bids of the other agents, then it should be so for all bids of the other agents. If this is the intention, it would be good to clarify.
- The section on additional related work in the body of the paper is very brief and not so informative. I guess that this may have been shortened due to the space limitation in the submission. Perhaps the authors could consider also deferring this paragraph to the appendix (or if there is space in the body of the paper, merging it with appendix A inside the paper as a full section).

**Questions:**

1)Combining competing ads: A concern that arises with the idea of merging content of different bidders, as in the example in page 1, is about cases when the bidders promote competing products, or more generally have competing economic interests. I could imagine how this can be a reasonable scenario, since if the bidders are competing for the same ad space (I.e., want their ad to be presented to the same consumers at the same times and locations, and, who perhaps were even searching for similar products), then the advertiser may be selling the same type of product.

An LLM will produce an output combining both ads even in cases when advertiser interests are conflicting (e.g., take a flight to Hawaii with firm-A-airlines and with firm-B-airlines). How can a mechanism such as the one proposed prevent such cases? Or more generally, assure that it creates ads that make sense in the market (at least are not economically self-contradictory)?

2)Selling an item that is different from the one that the buyers asked to buy: Continuing the previous point, in the classic model of auctions, as the paper also mentions, it is assumed that bidders want to buy the item that is being sold, which is modeled as some value that they have for it. In particular, for ads, the bidders have some value that they perceive for the right to use the ad space, and the auction model makes it is completely the responsibility of the bidders (who win the auction) to form their valuations. If I understand correctly, in the proposed framework, this is no longer the case. The advertisers wish to present some content, but then they may end up winning the auction and paying to the auctioneer for presenting a different content. Is there a way to verify that the buyer (bidder) actually wants to pay for the resulting product they buy? The first issue is with merging content, and a different and perhaps more subtle issue is that payments are set by the average of the output distribution where the actual output sampled for the ad might be far from some of the agents’ preferences.

As LLMs are black boxes, this seems challenging to have guarantees that their output makes sense economically (or makes sense at all), which is part of the reason this paper is interesting. I believe that some discussion of the above points may be useful for the paper.

3)Calibration of payments to actual willingness to pay: The payment rule suggested addresses the important point of making the payment aligned with ordinal preferences that the agents have (partially) over outcomes. However, eventually, the agents operate for some firms than need to pay when winning the auction. It is not clear how do these payments relate to actual monetary amounts that the owners of the agents are willing to pay. Specifically, if no values, budgets, or other monetary preferences were given to the agents, how can they calibrate their preferences over tokens to actual money?  It seems unlikely that the owner of the LLM will specify to their LLM agent their willingness to pay for every possible outcome or distribution over outcomes.

4)Stateless agents: In Section 1.2, first paragraph, the statement “One salient feature of the state-of-the-art LLMs is that they are stateless, i.e., they maintain no internal memory or state” is not entirely clear to me. I believe that the intention is that the models are trained offline and their neural network itself is then fixed. However, LLM sessions do have some form of a state which records the history of input and output tokens across different prompts – e.g., it is possible to ask an LLM to print again the first prompt in the current session, or to refer it to the last two prompts and generate a combined prompt and respond to it, etc. It would be good to clarify this part further. Does this kind of memory have implications on the framework? If so, it would be good to discuss them.

5)Related work: Appendix A does seem to describe the technically related work and provide some key pointers on mechanism design, learning, and auctions. The context of this paper is studying how mechanisms should operate when instead of classic players, the game is played by automated learning algorithms. This connects to a recent line of work that is currently missing from the discussion, which studies how incentives are generated for users of learning algorithms in various contexts, including in auctions (though using different models). I suggest adding some discussion of references along these lines.

[1] https://proceedings.neurips.cc/paper_files/paper/2022/file/b39fcf2e88dad4c38386b3af6edf88c7-Paper-Conference.pdf

[2] https://dl.acm.org/doi/pdf/10.1145/3485447.3512055

[3] https://arxiv.org/pdf/2307.07374.pdf

[4] https://dl.acm.org/doi/pdf/10.1145/3543507.3583416

**Ethics Review Description:**

The paper does theoretical modeling, there does not seem to be any ethical issue.

**Reviewer Confidence:**

3: The reviewer is confident but not certain that the evaluation is correct

**Scope:**

4: The work is relevant to the Web and to the track, and is of broad interest to the community

---

### Official Review · Reviewer_zwT4 · 2023-11-22

**Novelty:** 6
**Technical Quality:** 6

**Review:**

The authors consider the problem of designing auctions and mechanisms in an advertising setting with content to be generated from large language models.  More specifically, the authors propose a model which they call the “token auction” model wherein bidders specify public LLMs which are modeled as preferred distributions over tokens and submit bids to influence how closely a generated aggregate distribution over tokens resembles their desired distribution.  To this end, the authors define two natural incentive properties which essentially “pin down” the possible aggregation functions over the LLMs which allow for truthful implementation (akin to a Myersonian monotonicity condition in traditional auction settings).  They further use this characterization to define second-price-like payment rules.  Finally, the authors examine possible aggregation functions inspired by the training of LLMs and argue that an aggregation function inspired by KL divergence (i.e., a linear aggregation rule) - a loss function common in the first stage of LLM training -  is monotone whereas an alternative aggregation function inspired by RL-stage training (i.e., a log-linear aggregation rule) is non-monotone.  They ultimately evaluate their aggregation functions on a toy model with two LLM agents demonstrating that as one agent’s bid becomes gradually larger (relative to the other bid) the resulting aggregate token generation becomes gradually closer to the agent’s preferred token generation.

On the positive side, this paper feels very timely and introduces an interesting auction model for a natural emerging setting which is likely to be of interest to many in the WebConf community.  I agree with the sentiment in the paper that LLMs are likely to be an important part of the ad auction ecosystem, and I think this paper presents a nice “proof of concept” first step toward thinking about how one should design mechanisms in this new space.  Furthermore, the results in the paper, while not particularly technically demanding, in my opinion, are the “right” suite of results for an initial paper proposing a new line of larger open questions and, thus, “clear the bar” in my view.  On the negative side, although I like the model and results, I am not sure it captures some of the fundamental tradeoffs in this setting.  In particular, consider two competing firms offering a similar service.  A generated set of tokens which mentions both competing firms may not have positive value to either firm (due to the externality generated by the mention of the other).  I do not think such a scenario can neatly be captured in the proposed token auction framework in this paper.  However, I do not think oversights in the initial model the authors propose significantly detracts from the work, but I would suggest that the authors add discussion about extensions to, drawbacks of, and future questions regarding their proposed setting to paint a more complete picture.  On the whole, I am positive about this submission.

Lines 75-93: I would suggest using a consistent capitalization/case choice for “Maui Airlines” and “Stingray Resort”

Line 303: I wonder if “obvious” is the right word to use here (and elsewhere).  Perhaps “natural” is better?  I don’t think it is too significant, but “obvious preferences”/”obvious strategyproofness” now have strong behavioral game theory connotations (see, e.g., [Zhang and Levin 2017] “Partition Obvious Preference and Mechanism Design: Theory and Experiment”)

Line 839: “we need to peak” -> “we need to peek”

Line 846: “we start with a based model” -> “we start with a base model”


[After rebuttal] I thank the authors for their responses to my questions as well as the questions of the other reviewers.  I am positive about this paper and would recommend acceptance.

**Questions:**

The objectives of the bidders (and welfare function of the central planner) that you propose seem to share some common "spirit" with the literature on truthful aggregation of budget proposals (see, e.g., [Freeman et al. 2021] in the Journal of Economic Theory).  Can you comment on whether insights from that literature have any relevance to your setting (or vice-versa)?

**Reviewer Confidence:**

3: The reviewer is confident but not certain that the evaluation is correct

**Scope:**

4: The work is relevant to the Web and to the track, and is of broad interest to the community

---

### Official Review · Reviewer_7WDb · 2023-11-23

**Novelty:** 7
**Technical Quality:** 6

**Review:**

## After rebuttal

After reading the author's response and the other reviews, I still believe this is a solid paper for the conference.

## Summary:

The paper model agents that have preferences over content created by large language models (LLMs) and propose a proper auction design to aggregate the created content together, for example, to create an aggregated advertisement for various products or services in a dialog of a famous video game.  They demonstrate how to design an auction in the spirit of the well-known "second-price auction" and propose aggregation functions for this process.

## Evaluation:

The paper proposes a new model, essentially combining mechanism design with a topic of extreme interest. Crucially, the connection is not straightforward.

### Pros:

1. The paper works on an area of extreme interest and makes the first connection of mechanism design techniques to the area.
2. The paper makes a first and quite successful attempt to model preferences for LLM agents.
3. The paper is well-written and technically sound.

### Cons:

I could not identify any major concerns.

**Questions:**

1. While the obvious preferences modeling seems compelling for LLM agents, have you considered any alternatives? It would be nice to see any modeling examples that didn't work well or some high-level thoughts on possible future directions.

2. Regarding the necessity of randomization in line 109: Can you add some indicative citations or a quick explanation for the interested reader to be able to follow up with this?

**Ethics Review Description:**

-

**Reviewer Confidence:**

3: The reviewer is confident but not certain that the evaluation is correct

**Scope:**

4: The work is relevant to the Web and to the track, and is of broad interest to the community

---

### Official Review · Reviewer_mj3d · 2023-11-24

**Novelty:** 6
**Technical Quality:** 5

**Review:**

The authors study a mechanism design problem for large language models, that is motivated by the fact that AI-generated ads can combine input from different advertisers. They formulate the problem based on a token auction model, and they design two incentive properties that end up to be equivalent to monotonicity conditions on distribution aggregation. They then show that  for such aggregation functions it is possible to design second price auctions despite the fact that  in this model there are no bidder valuation functions. Finally, they design specific aggregation functions and provide analysis of the auction both theoretically and experimentally.

Although I am not that familiar with the area, I liked the problem that the paper studies, I found it well-motivated, and in general I think that it is well-written and manages to convey the message even to the unfamiliar reader. The formulation that the authors propose is concrete and the analysis of the model is also nice and involved. I do not have any major complaints apart form the fact that the complexity of the model makes some parts hard to evaluate. Overall, I would say that this is an interesting paper that presents a nice collection of results and probably is a good match for the conference.

**Questions:**

None.

**Reviewer Confidence:**

2: The reviewer is willing to defend the evaluation, but it is likely that the reviewer did not understand parts of the paper

**Scope:**

3: The work is somewhat relevant to the Web and to the track, and is of narrow interest to a sub-community

---

### Official Review · Reviewer_rLpi · 2023-11-27

**Novelty:** 6
**Technical Quality:** 4

**Review:**

This paper explores the problem of designing auction mechanisms to support the emerging format of AI-generated content, with a focus on aggregating several Large Language Models (LLMs) in an incentive compatible manner. The authors propose a general formalism called the token auction model for studying this problem.

The paper first proposes a robust auction design approach that assumes agent preferences entail partial orders over outcome distributions. The authors formulate two natural incentive properties and show that these are equivalent to a monotonicity condition on distribution aggregation. They also show that for such aggregation functions, it is possible to design a second-price auction, despite the absence of bidder valuation functions.

The authors then design concrete aggregation functions by focusing on specific valuation forms based on KL-divergence, a commonly used loss function in LLM. The welfare-maximizing aggregation rules turn out to be the weighted (log-space) convex combination of the target distributions from all participants.

**Questions:**

This paper considers a mechanism design problem from an innovative perspective and it seems that the model can be applied to realistic scenarios. Even though part of the results focuses on the property characterization of mechanisms, it is the first paper to take LLM into mechanism design which may stipulate subsequent work.

Can the authors give more explanation on the utilities of advertisers and how the bids affect these utilities?

**Reviewer Confidence:**

2: The reviewer is willing to defend the evaluation, but it is likely that the reviewer did not understand parts of the paper

**Scope:**

3: The work is somewhat relevant to the Web and to the track, and is of narrow interest to a sub-community

---

### Decision · Program_Chairs · 2024-01-22

**Decision:**

Accept (Oral)

**Comment:**

This paper studies how to design auction mechanisms for ads that are generated by LLMs, and bidders can submit both LLMs as well as bids.
 It proposes a novel model for this setting, a novel auction format generalizing second-price auctions for this model, and analyze the proposed auction theoretically and with experiments.

 The review team identified the following strengths and weaknesses of the submission:

 Strengths:

 - Novel, timely, and innovative mechanism design problem that is likely to be of broad interest to the WebConf community

 - The proposed model is non-trivial, original, and makes sense for this novel setting.

 - The technical results are natural, sound and seem to be the 'right' results for this new model

 Weaknesses:

 - There are some clear extensions of the model (e.g. dealing with substitutes) that are of first order of interest for the settings described in the paper, but which are not addressed by the authors in the submission

 Overall, the review team unanimously finds the paper novel, well-executed, and likely to be of broad interest. It has potential to be a landmark paper sparking a new line of research linking LLMs and mechanism design. I recommend to accept the paper.